# HGF-Induced PD-L1 Expression in Head and Neck Cancer: Preclinical and Clinical Findings

**DOI:** 10.3390/ijms21228770

**Published:** 2020-11-20

**Authors:** Verena Boschert, Jonas Teusch, Anwar Aljasem, Philipp Schmucker, Nicola Klenk, Anton Straub, Max Bittrich, Axel Seher, Christian Linz, Urs D. A. Müller-Richter, Stefan Hartmann

**Affiliations:** 1Department of Oral and Maxillofacial Plastic Surgery, University Hospital Würzburg, D-97070 Würzburg, Germany; teusch.jon@googlemail.com (J.T.); aljasem.anwar@gmail.com (A.A.); zahnaerztin.lh@gmail.com (N.K.); Straub_A@ukw.de (A.S.); Seher_A@ukw.de (A.S.); linz_c@ukw.de (C.L.); Mueller_U2@ukw.de (U.D.A.M.-R.); 2Comprehensive Cancer Center Mainfranken, ECTU, University Hospital Würzburg, D-97080 Würzburg, Germany; philipp.schmucker@stud-mail.uni-wuerzburg.de; 3Department of Internal Medicine II, University Hospital Würzburg, D-97080 Würzburg, Germany; Bittrich_M@ukw.de

**Keywords:** HNSCC, head and neck cancer, HGF, Met, PD-L1, immune therapy

## Abstract

Head and neck squamous cell carcinoma (HNSCC) is a widespread disease with a low survival rate and a high risk of recurrence. Nowadays, immune checkpoint inhibitor (ICI) treatment is approved for HNSCC as a first-line treatment in recurrent and metastatic disease. ICI treatment yields a clear survival benefit, but overall response rates are still unsatisfactory. As shown in different cancer models, hepatocyte growth factor/mesenchymal–epithelial transition (HGF/Met) signaling contributes to an immunosuppressive microenvironment. Therefore, we investigated the relationship between HGF and programmed cell death protein 1 (PD-L1) expression in HNSCC cell lines. The preclinical data show a robust PD-L1 induction upon HGF stimulation. Further analysis revealed that the HGF-mediated upregulation of PD-L1 is MAP kinase-dependent. We then hypothesized that serum levels of HGF and soluble programmed cell death protein 1 (sPD-L1) could be potential markers of ICI treatment failure. Thus, we determined serum levels of these proteins in 20 HNSCC patients before ICI treatment and correlated them with treatment outcomes. Importantly, the clinical data showed a positive correlation of both serum proteins (HGF and sPD-L1) in HNSCC patient’s sera. Moreover, the serum concentration of sPD-L1 was significantly higher in ICI non-responsive patients. Our findings indicate a potential role for sPD-L1 as a prognostic marker for ICI treatment in HNSCC.

## 1. Introduction

Recently, immune checkpoint inhibitors (ICIs) were outstandingly successful in the treatment of cancer. This new form of immunotherapy emerged from the discovery that, apart from the costimulatory receptors expressed on antigen-presenting cells, receptors also exist that deliver inhibitory signals to T-lymphocytes. Like the costimulatory receptors that are mandatory for the activation of T-lymphocytes, they help to constrain self-reactive T-lymphocytes and, therefore, are important checkpoints regulating the balance between self-tolerance and autoimmunity [1]. One of these checkpoints is PD-1 (programmed cell death protein 1) with its ligands PD-L1 and PD-L2 [1,2,3]. Apart from several types of immune cells and different non-haematopoetic cells, tumor cells have been found to express PD-1 ligands as well. When being recognized by T-lymphocytes via binding of T-cell receptor and tumor antigen, an efficient T-cell-activation is prevented by this inhibitory immune checkpoint, and, therefore, the tumor cell is able to escape from the immune system.

Antibodies preventing the interaction of checkpoint receptors with their ligands have shown great success in the treatment of different cancers, such as melanoma [4], non-small-cell lung carcinoma (NSCLC) [5], or Hodgkin’s lymphoma, in recent years [6]. In 2019, both the US Food and Drug Administration and the European Commission granted approval for PD-1 inhibition as a first-line treatment for patients with metastatic or unresectable, recurrent head and neck squamous cell carcinoma (HNSCC) as monotherapy or in combination with platinum and 5-fluorouracil (5-FU) chemotherapy. A randomized, international phase 3 clinical trial of participants with untreated locally incurable recurrent or metastatic HNSCC found that the ICI pembrolizumab, a neutralizing PD-1 specific antibody, is an appropriate first-line treatment for this indication (KEYNOTE-048, NCT02358031) [7].

In 2018, ~835,000 new cases of head and neck cancer were reported worldwide [8]. Major risk factors for developing this disease are consuming tobacco, alcohol abuse, poor oral hygiene, and human papillomavirus (HPV) infection. Survival is low: the EUROCARE-5 study reported a 5-year survival rate of 40% in Europe in the early 2000s (cases of the larynx not included) [9]. More current data of the years 2016–2018 show a 5-year survival rate of 55% for cases located at the oral cavity or the pharynx in Germany (www.krebsdaten.de). Squamous cell carcinomas are the most abundant form of head and neck cancer, reported to comprise 90% of all cases [10]. Until recently, the only available targeted therapy for HNSCC was cetuximab, a monoclonal antibody targeting the epidermal growth factor receptor (EGFR) [11]. Treatment with cetuximab in combination with radiation or chemotherapy results in a significant survival benefit compared to standard therapy alone, but unfortunately, a lot of tumors also develop resistance against this antibody [12]. The new PD-1 specific antibody pembrolizumab showed a significantly higher survival benefit compared to cetuximab treatment (13.6 vs. 10.4 months, combined with chemotherapy, PD-L1 positive participants) [7]. Nevertheless, the overall response rates (ORR) for immune checkpoint inhibition in HNSCC are low. For the KEYNOTE-048 trial, an ORR of 36.4% was reported for PD-L1 positive participants receiving pembrolizumab plus chemotherapy (NCT02358031) [7]. Several reasons for possessing or acquiring resistance against ICIs are proposed, either on the side of the tumor cell (e.g., by malfunctioning of the antigen-presenting machinery) or the T-cell (e.g., by mutations in the interferon-γ-regulating genes) [13].

Some cytokines and growth factors have been reported to increase the concentration of checkpoint ligand PD-L1 on tumor cells, such as interferon-γ (IFNγ) [14], tumor necrosis factor α (TNFα) [15], or epidermal growth factor (EGF) [16]. In addition, hepatocyte growth factor (HGF) increases PD-L1 concentration in different tumor cell lines [17,18]. This growth factor signals by binding to its receptor Met (mesenchymal–epithelial transition), representing a receptor tyrosine kinase (RTK) overexpressed in 80% of HNSCCs [19]. Furthermore, HGF/Met signaling was found to be implicated in cetuximab resistance [20,21].

In light of the small response rates towards ICI-treatment, there is a need for biomarkers that make it possible to predict how patients respond to this treatment [13]. Interestingly, HGF concentration in serum was found to be higher in the later stages of metastatic melanoma [22]. Moreover, melanoma-patients treated with ICIs and with high HGF serum concentrations were more likely to react with no response upon therapy [22]. These findings indicate that HGF serum concentration could represent a predictive marker for the success of a therapy with checkpoint inhibitors.

By proteolytic cleavage of membrane-bound PD-L1, a soluble form can arise (sPD-L1), which is detectable in human serum [23,24]. In serum of patients suffering from different types of cancer, this form of the checkpoint ligand has been detected so far and found to be a potential prognostic or therapeutic marker [25,26,27]. Whether a high level of sPD-L1 in serum correlates with a positive or negative outcome cannot be generalized, as results differ between investigations and tumor entities [24].

In our investigation, we wanted to determine if the effect of HGF on PD-L1 concentration is also detectable in HNSCC. Therefore, we examined the PD-L1 mRNA and protein concentration in HGF stimulated HNSCC cell lines of different origins and verified if the detected changes are really specific for HGF/Met signaling. Furthermore, we investigated if the HGF and sPD-L1 concentrations in the serum of HNSCC patients differ between different outcomes of ICI treatment and if the concentrations of the two proteins correlate with each other.

## 2. Results

For investigating the effects of HGF stimulation in HNSCC on PD-L1 concentration, we chose three commercially available cell lines: Detroit 562, FaDu, and SCC-9. We already used these three cell lines in a project dealing with glycolytic reprogramming upon HGF stimulation [28]. As part of this project, RNA was isolated, and mRNA sequencing was performed after stimulating the cells with 50 ng/mL HGF for 16 h. Figure 1a shows the results of this mRNA sequencing for the levels of the mRNA of CD274, the gene coding for PD-L1. Actually, in all three cell lines, expression levels of CD274 were higher upon HGF stimulation, especially in Detroit 562, where this increase showed an adjusted *p*-value low enough to be significant. When we examined the expression levels using qPCR, we also found an increase in expression upon HGF stimulation (Figure 1b). We wondered if higher levels of PD-L1 could also be detected on the protein level, so we performed Western blot analysis with lysates of cells stimulated with HGF for 48 h using an antibody specific for human PD-L1. As a negative control, we additionally treated cells with 0.5 µM of the Met-specific tyrosine kinase inhibitor, foretinib. One can see for all three cell lines that the protein band for PD-L1 was more intense when cells were treated with HGF (Figure 1c). In the presence of HGF and foretinib, the intensity of the protein band was comparable to control, implying a Met specificity of the increase in PD-L1 protein.

To check if this higher level of protein also leads to a higher level of PD-L1 protein presented on the cell surface, we performed flow cytometry using an allophycocyanin (APC)- conjugated PD-L1 antibody recognizing the extracellular part of the immune checkpoint protein-ligand (Figure 2). Upon HGF stimulation, one can clearly see a shift in the histograms to higher fluorescence intensities compared to the untreated controls in all three cell lines (Figure 2a, orange compared to red). In the presence of foretinib, results were similar to the ones of the untreated samples (Figure 2a, blue compared to red). Summing up several experiments using the median of fluorescence illustrates the effect of HGF: fluorescence intensities were significantly increased upon HGF stimulation compared to the untreated control, and the addition of foretinib was able to prevent this increase (Figure 2b).

To further confirm the Met-specificity of the enhanced surface expression of PD-L1 upon HGF stimulation, we transfected Detroit 562 cells using two different Met specific siRNA constructs (Met siRNA I or Met siRNA II) or an unspecific control siRNA. Two days after transfection, cells were stimulated with HGF or remained untreated for 48 h. Subsequently, cells were stained with APC coupled PD-L1 specific antibody or with a phycoerythrin (PE)-conjugated antibody specific for the extracellular part of Met. Detroit 562 cells express high levels of Met receptor, as we already know from previous experiments [28]. Accordingly, the sample transfected with control siRNA showed a strong PE fluorescence when stained for Met receptor (Figure 3a,b in violet). Stimulation with HGF led to a small increase in Met receptor surface expression in Detroit 562 cells transfected with control siRNA (Figure 3a,b in green). When cells were transfected with Met siRNA I, a shift towards lower fluorescence intensities can be seen, regardless whether stimulated or not (Figure 3b,c). This shows that Met expression was successfully silenced, and Met receptor concentration on the cell surface was low after transfection. Transfecting the cells with Met siRNA II also led to a successful silencing, although a bit more Met receptor remained on the cell surface after transfection compared to cells treated with siRNA I (Figure 3c).

In cells transfected with control siRNA, the expected increase in PD-L1 protein could be seen upon HGF stimulation (Figure 3d,e, red compared to blue). However, in HGF stimulated cells transfected with Met siRNA I, this increase was much smaller (Figure 3e), and median fluorescence was much lower (Figure 3f). Transfection with Met siRNA II was also able to prevent the increase in PD-L1 protein in part, but with lower efficacy compared to Met siRNA I (Figure 3e). Experiments with FaDu cells had a comparable outcome (Appendix A). The results of the siRNA experiments, therefore, confirmed the Met-specificity of the observed increase in PD-L1 upon HGF treatment in HNSCC-cells.

The main signaling events that become activated after HGF-binding to Met are mediated by PI3K/Akt/mTOR, Ras/Raf (MAP kinase signaling pathway), and Stat3. To determine which of these three pathways are necessary for the increase in PD-L1 protein, we applied an inhibitor for Stat3 (S3I-201), Akt (MK-2206), and the MAP kinases Erk1/2 (SCH772984) to Detroit 562 cells stimulated with HGF. Looking for changes in PD-L1 concentration using Western blot analysis, we found that with SCH772984, the MAP kinase inhibitor, PD-L1 concentration was not increased upon HGF stimulation (Figure 4a). To further confirm this, we applied an additional inhibitor, trametinib, which inhibits Mek1/2, the kinases phosphorylating Erk1/2 in the MAP kinase cascade. Western blot analysis showed that trametinib was able to prevent an increase in all three cell lines (Figure 4b). To further confirm the involvement of the MAP kinase pathway, we applied siRNA specific for Erk1/2 (Figure 4c). As expected, when cells were transfected with a control siRNA, PD-L1 concentration increased when cells were treated with HGF. With Erk1/2 siRNA, this increase was comparably lower in all three treated cell lines (Figure 4c). Taken together, these findings imply that the MAP kinase signaling pathway is responsible for the changes in PD-L1 concentration upon HGF stimulation that we observed in HNSCC cell lines.

To further validate our preclinical findings, we evaluated serum HGF and sPD-L1 levels from peripheral blood samples of HNSCC patients before ICI treatment. All patients were in the recurrent and/or metastatic stage of the disease. Further information on the localization of the tumors and initial treatment can be found in the Materials and Methods section. A total of 20 sera were available and included in the study. The cohort consisted of 4 women and 16 men. The mean age of the cohort was 62.3 years. All patients were treated with ICIs specific to the PD-1/PD-L1 checkpoint. Nine patients received nivolumab monotherapy, seven patients received a combination of nivolumab and ipililumab, three received a combination of nivolumab and a GITR antibody, one received durvalumab. As illustrated in Figure 5a, HGF and sPD-L1 serum levels showed a positive and significant correlation in these patients (*p* = 0.014). Spearman’s r was 0.5398, indicating a strong correlation in the context of biomedical data [29]. We then went on to analyze serum levels of HGF and sPD-L1 in view of the clinical response to ICI treatment (Figure 5b,c). Therefore, we defined patients with complete remission (CR), partial remission (PR), and stable disease (SD) as responders and patients with progressive disease (PD) and death during therapy as non-responders. For both tested parameters, we observed higher serum levels in non-responders. HGF serum level in responders was 291.4 pg/mL compared to 371.3 pg/mL in non-responders (Figure 5b). However, this finding showed a trend but was not significant. Mean serum level of sPD-L1 was 74.02 pg/mL in the responder group and 94.76 pg/mL in the non-responder group (Figure 5c). In contrast to HGF, this difference in sPD-L1 concentration between responders and non-responders was significant (*p* = 0.0201).

## 3. Discussion

HGF/Met signaling contributes to metastasis, proliferation, anti-apoptotic signaling, and migration in HNSCC [30]. Accordingly, Met was found to be overexpressed in a high percentage of HNSCC tumor samples (Met^high^ tumors) [19]. Additionally, HGF/Met signaling seems to be involved in the immunosuppression of tumors [31]. In light of the recent approval of ICIs for HNSCC as a first-line treatment in recurrent and metastatic disease, it is of interest to learn more about the connections between HGF/Met and immune checkpoints in HNSCC. Hence, we aimed to investigate the influence of HGF/MET signaling on the expression level of the immune checkpoint protein PD-L1 in HNSCC.

In three HNSCC cell lines, we could determine that HGF stimulation can lead to higher levels of PD-L1 on mRNA and the protein level. Noteworthy, the cell surface-located proportion of the PD-L1 protein was also significantly enhanced upon HFG stimulation. These effects were specific for Met, as inhibiting the receptor using the Met-specific tyrosine kinase inhibitor foretinib or degrading Met-mRNA using specific siRNAs for Met impeded the increase. Furthermore, we could show that for this effect, the MAP kinase signaling pathway is necessary, as two chemical inhibitors of MAP kinase phosphorylation were able to prevent the increase in PD-L1 protein. Additionally, HGF stimulation of cells transfected with Erk1/2 specific siRNA resulted in a less prominent increase.

In renal cancer cells, it has been shown that stimulation with HGF increases PD-L1 levels via the MAP kinase pathway [17]. Furthermore, Ahn et al. detected an increase upon HGF stimulation in a lung adenosquamous cancer cell line (H596) [18]. In Met-amplified cell lines (H596 and HS746T), treatment with Met-specific tyrosine kinase inhibitors reduced PD-L1 levels. The latter was also discovered in another study using several Met amplified tumor cell lines [32]. Furthermore, this investigation showed that PD-L1 increase upon IFNγ stimulation was also impaired when cells were treated with Met-inhibitors. Interestingly, in liver cancer, the situation seems to be different, as treatment with Met-inhibitors leads to an increase in PD-L1 rather than a decrease [33]. This shows that there can be differences in PD-L1 regulation between cancer diseases.

Investigations trying to determine the influence of Met on PD-L1 in other tumor diseases already found a high level of PD-L1 in Met amplified cell lines without HGF treatment [18,33]. In our study, we investigated the Met amplified cell line Detroit-562 [28]. It did not harbor significantly more PD-L1 mRNA or protein than the other two cell lines without HGF stimulation (see Figure 1a,c). This shows that Met amplification alone does not generally result in a high PD-L1 concentration. Control and feedback mechanisms in the cell and probably also the necessity for a certain amount of receptor molecules presented on the cell surface could be the cause for differences between Met-amplified cell lines.

To gain insight into the clinical importance of our cell line data, we determined sPD-L1 and HGF serum levels in HNSCC patients before they were treated with different ICIs specific for the PD1/PD-L1 checkpoint. We wondered if, due to the connection between HGF and PD-L1 found in the cell lines, we could detect a positive correlation between HGF and sPD-L1. Furthermore, HGF and/or sPD-L1 concentration in serum before treatment could correlate with the clinical outcome of the treatment. Indeed, we found a positive correlation between HGF and sPD-L1 and higher levels of the proteins in cases of progressive disease and death. For sPD-L1, this enhanced protein level was significant when compared to non-responders.

HGF levels in serum have been reported to be of diagnostic or prognostic value in different cancers by several investigators [34]. In HNSCC, Kim et al. showed that the HGF-levels in serum significantly correlated with tumor stage progression [35]. Druzgal et al. determined serum concentrations of different serum cytokines before and after treatment of HNSCC patients [36]. They found that HGF levels were significantly increased in patients compared to healthy volunteers, and no association with overall survival or disease-free survival after treatment (chemotherapy, radiation, and/or surgery) could be found. Only an increase in post-treatment levels of HGF was determined to be associated with poor survival. In contrast, in a study looking at HGF serum levels ahead of nivolumab or pembrolizumab treatment of patients suffering from metastatic melanoma, non-responders showed significantly higher serum levels than responders [22]. In our investigation, we found, on average, a higher HGF-level in patients not responding to ICI treatment as well, but the difference in the levels of the responders was not significant. For further corroboration of these findings, investigating a larger cohort of HNSCC patients treated with ICIs is necessary. Furthermore, looking at HGF-concentration after ICI treatment could be reasonable, as it could be of prognostic value to predict a recurrent disease.

sPD-L1 is a soluble form of PD-L1 which is detectable in human serum. Whether it fulfills an active function is still a matter of debate [24]. Metalloproteases cleave the membrane form of the protein, thereby generating its soluble form [37]. Another possibility for its formation is the existence of differently spliced mRNAs of PD-L1 [38,39]. Many studies have been published that investigate the clinical impact of sPD-L1 [24]. Whereas some found a high level of soluble checkpoint ligand to be of benefit for treatment [40], others declared it to be an indication for poor prognosis and shorter survival times [41,42]. Mazzaschi et al. specifically investigated sPD-L1 concentration in the context of ICI treatment using sera of NSCLC patients [43]. They showed that high sPD-L1 concentrations, together with low numbers of CD8^+^ and PD-1^+^ immune cells and NK-cells, have a negative impact on progression-free survival. Our investigation of HNSCC patients treated with ICIs supports these findings: Patients that did not respond to treatment showed significantly higher sPD-L1 serum levels before treatment. Furthermore, the positive correlation of sPD-L1 and HGF serum levels we found could imply that HGF/Met signaling contributes to high sPD-L1 levels. HGF could not only enhance mRNA of the full-length protein but also of its splice variants composed solely of the extracellular protein part. Likewise, higher levels of PD-L1 on the cell surface could result in higher levels of protein processed by metalloproteases. However, until now, it has not been known what regulates the processing of PD-L1 by the proteases.

With regard to the link between HGF/Met signaling and PD-L1 shown in this and several other publications, combination therapy of ICIs targeting PD-1/PD-L1 with a therapy targeting HGF/Met could be an option for ICI-refractory and Met^high^ patients. However, no therapy for HNSCC targeting HGF/Met signaling is approved so far, although the pathway is often found to be activated in this disease [30].

ICIs targeting PD-1 and PD-L1 are a great benefit for the treatment of HNSCC patients, especially when bearing in mind that targeting therapies for this disease are scarce. Nevertheless, the percentage of patients responding to this immunotherapy still is low. To establish ways of finding the patients that will benefit from treatment could be one building block in enhancing response rates. Another could be the combination with a different therapy. HGF/Met signaling could be an interesting point of attack, but further investigations are needed.

## 4. Materials and Methods

### 4.1. Cell Lines, Inhibitors, and Antibodies

HNSCC cell lines Detroit 562, FaDu, and SCC-9 were obtained from ATCC (Manassas, VA USA).

The following inhibitors from Selleckchem (Houston, TX, USA) were used as follows: foretinib (0.5 µM), MK-2206 (1 µM), S3I-201 (100 µM), SCH772984 (1 µM), and trametinib (25 nM).

Antibodies for PD-L1 (13684), Erk1/2 (9107), P-Erk1/2 (4370), Vinculin (13901), Akt (4691), P-Akt (4060), Stat3 (9139), and P-Stat3 (9145) were obtained from Cell Signaling (Danvers, MA, USA). APC conjugated PD-L1 antibody (17-5983-41), and tubulin-antibody (MS-481-P) were purchased from ThermoFisher Scientific (Waltham, MA, USA), β-Actin antibody (A5316) from Merck, and PE conjugated Met antibody (FAB3582P) from R&D Systems (Minneapolis, MN, USA). All antibodies were applied as stated in the instructions of the manufacturers.

### 4.2. mRNA Sequencing

Cells were seeded in 6-well plates (750,000 cells/well). On the next day, cells were treated in triplicates with 50 ng/mL HGF or were left untreated. Treatment lasted for 16 h, and then cells were washed with 2 mL cold PBS and detached from the plate using 1 mL of 10% Trypsin-EDTA (Merck, Darmstadt, Germany). After spinning down cells for 5 min at 2000 rpm and 4 °C, total RNA was isolated using a commercially available kit (RNeasy Mini Kit, Qiagen, Hilden, Germany), as described before. The detailed protocol for mRNA sequencing is described elsewhere [28]. The accession number for the raw data is GSE135552 at the Gene Expression Omnibus [44].

### 4.3. qPCR

Cells were treated as described in Section 4.1. After RNA isolation (see Section 4.2), cDNA was generated using the QuantiTect Reverse Transcription Kit (Qiagen, Hilden, Germany) in accordance with the instructions provided by the manufacturer. In the PCR reaction, 20 ng of cDNA was used with 1.5 µL of PD-L1 Primer (QuantiTect Primer Assay, Qiagen) and 12.5 µL of a ready-to-use qPCR master mix (QuantiTect SYBR Green PCR Kit, Qiagen). qPCR and analysis of results were performed as described [28].

### 4.4. Western Blotting

Western blotting was performed as reported in [28]. Briefly, cells were treated for the indicated periods with 50 ng/mL HGF, with 50 ng/mL HGF plus the indicated inhibitors, or remained untreated. Samples for SDS-PAGE were made as described [28] and boiled for 5 min at 95 °C before loading. After gels were run and blotted, membranes were blocked with 5% dry milk in TBS for 1 h. Blots were incubated overnight at 4 °C with a primary antibody. The next day, blots were washed and incubated with the appropriate HRP-coupled secondary antibody for 1 h and, after washing, were subjected to signal detection (ECL Western Substrate, ThermoFisher Scientific, Waltham, MA, USA; ChemiDoc Imaging System, Bio-Rad, Hercules, CA, USA). Subsequently and if necessary, blots were incubated for 10 min in stripping buffer (Carl Roth, Karlsruhe, Germany), washed 5 times for 5 min with PBS, and treated as stated above for detecting a different protein.

### 4.5. Flow Cytometry

For detecting PD-L1 and Met receptor, HNSCC cells were seeded and treated as stated above in 4.2. They were detached from the plate using 10% Trypsin-EDTA after 48 h of stimulation and spun down at 600 g for 5 min at 4 °C. Cells were stained for 30 min with 2.5 µL per sample of APC coupled PD-L1 antibody or PE coupled Met antibody or with corresponding isotype control antibodies in PBS with 0.5% BSA. After incubation, samples were washed two times with PBS containing 0.5% BSA and were measured in a flow cytometer (BD FACSCalibur, BD Bioscience, San Jose, CA, USA). Results were analyzed using Flowjo software (BD Bioscience).

### 4.6. siRNA Transfection

A total of 625,000 HNSCC cells per sample were reverse transfected with 10 pmol Met siRNA I, Met siRNA II, or control siRNA (Signal Silence siRNA 6568, 6618 or 6622, Cell Signaling, Denvers, CO, USA). Per sample, 3 µL Lipofectamine RNAiMAX reagent (ThermoFisher Scientific, Waltham, MA, USA) was used. Further processing was performed according to the instructions provided by the manufacturer of the transfection reagent. For Erk1/2 siRNA (Signal Silence siRNA 6560, Cell Signaling), 20 pmol and 6 µL transfection reagent were used. Forty-eight hours after transfection, cells were stimulated with 50 ng/mL HGF or remained untreated. After 48 h of stimulation, cells were stained for receptor expression and analyzed using flow cytometry, as stated in Section 4.5. In the case of Erk1/2 silencing, cells were lysed and analyzed using Western blot, as stated in Section 4.4 after 48 h.

### 4.7. ELISA Measurements

Commercially available kits were used for ELISAs detecting PD-L1 (DB7H19, R&D systems, Minneapolis, MN, USA) and HGF (ELH-HGF, Ray Biotech, Peach Tree Corners, GA, USA). Experiments were implemented as stated in the instructions provided by the manufacturers.

### 4.8. Patients

The study included 20 patients with recurrent and/or metastatic HNSCC with an age range of 41–74 years (mean age: 62.3 years). Four patients were female, 16 patients were male. The primary tumors of 13 patients were located in the oropharynx, of three patients in the hypopharynx, and of two in the epipharynx. The tumor of one patient was located in the larynx, and one patient suffered from a tumor located on the floor of the mouth. Before the start of ICI-treatment, sixteen patients had a tumor resection and had been treated with adjuvant chemo- and/or radiotherapy, four patients had been treated solely with chemo- and radiotherapy, and two patients had only a tumor resection. All patients were treated at the University Hospital of Würzburg, Germany. The patients underwent clinical and/or radiological assessment three months after ICI treatment initiation. Patients with CR, PR, and SD were defined as responders, and patients with PD and death during therapy as non-responders. Based on this definition, among the 20 included patients, 8 patients responded to ICI treatment, whereas the other 12 patients did not. All subjects gave their informed consent for inclusion before they participated in the study. The study was conducted in accordance with the Declaration of Helsinki, and the protocol was approved by the Ethics Committee of the University Hospital Würzburg (No 15/18-me).

## Figures and Tables

**Figure 1 ijms-21-08770-f001:**
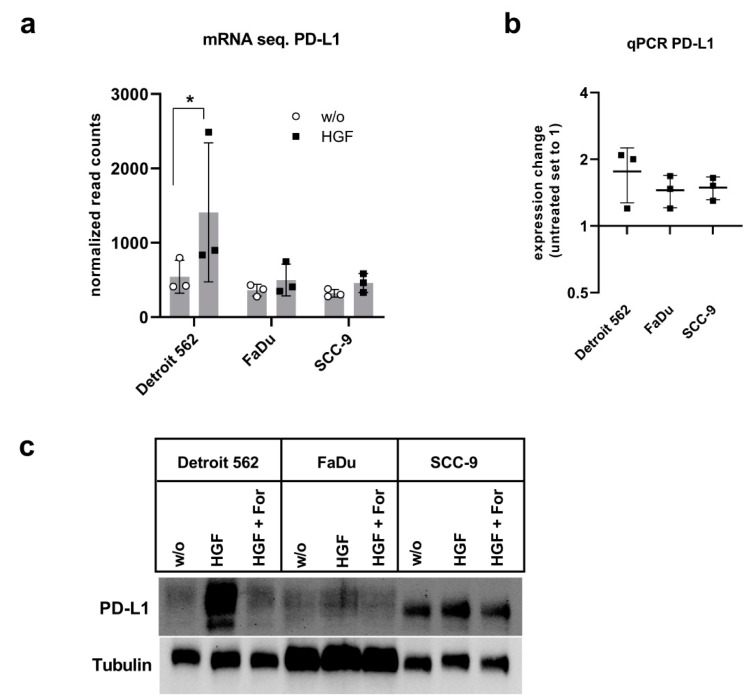
Hepatocyte growth factor (HGF) stimulation enhances programmed cell death protein 1 (PD-L1) concentration on mRNA and protein level. Three head and neck squamous cell carcinoma (HNSCC) cell lines (Detroit 562, FaDu, and SCC-9) were treated with 50 ng/mL HGF (HGF), 50 ng/mL HGF in combination with 0.5 µM foretinib (HGF + For) or remained untreated. After 16 h, RNA was isolated and subjected to mRNA sequencing (**a**) or was further processed to cDNA for qPCR (**b**). Additionally, cells were lysed after 48 h, and a Western blot with an antibody specific for PD-L1 was carried out (**c**). As a loading control blot was reprobed using an anti-tubulin antibody. *: adjusted *p*-value (from sequencing analysis) of 0.001 (α is 0.1).

**Figure 2 ijms-21-08770-f002:**
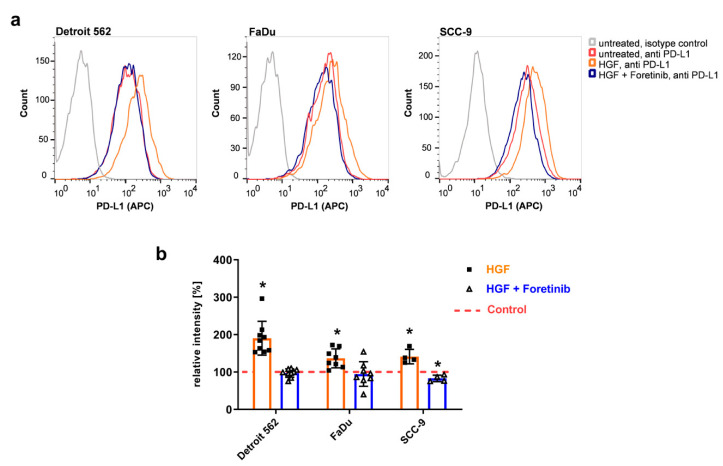
PD-L1 concentration on the cell surface is enhanced upon HGF treatment. Three HNSCC cell lines (Detroit 562, FaDu, and SCC-9) were treated with 50 ng/mL HGF (orange), HGF and 0.5 µM foretinib (blue) or remained untreated (red). After 48 h, cells were stained for flow cytometry using an APC-coupled antibody specific for PD-L1 or an isotype control (exemplarily shown for untreated control in grey). Histograms for one experiment out of at least four are shown in (**a**). A summary of all experiments shows fluorescence median values of all tested cell lines and conditions (corresponding isotype controls subtracted) relative to the untreated control, which is indicated by the dashed red line (**b**). *: *p*-value ≤ 0.05 in a one-sample *t*-test.

**Figure 3 ijms-21-08770-f003:**
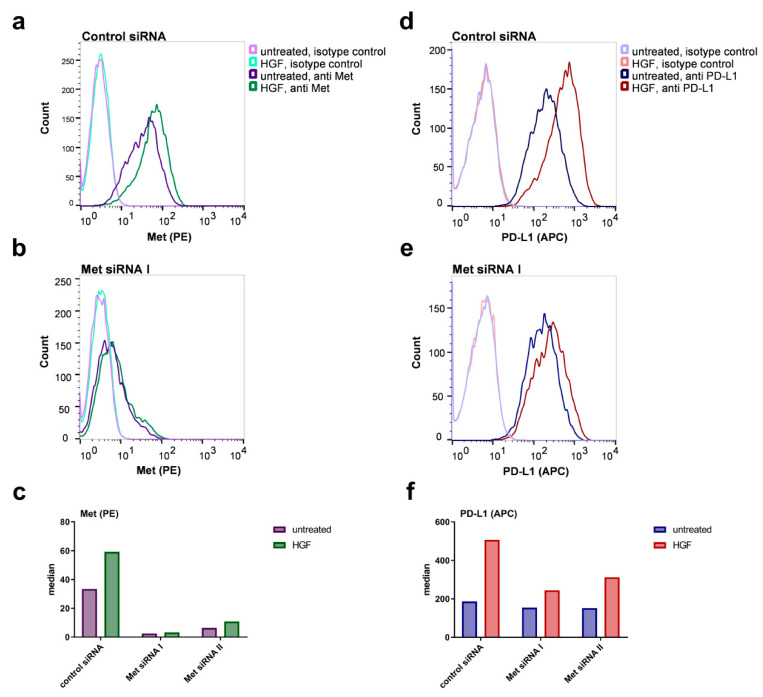
Increased PD-L1 concentration on the cell surface is mesenchymal–epithelial transition (Met)-receptor-dependent. Detroit 562 cells were transfected with two different siRNA constructs specific for the Met receptor (Met siRNA I and II) or a control siRNA. Forty-eight hours after transfection, cells were treated with 50 ng/mL HGF or remained untreated. Cells were subjected to flow cytometry after an additional 48 h using a phycoerythrin (PE)-conjugated Met specific antibody in (**a**–**c**), an allophycocyanin (APC)-conjugated PD-L1 specific antibody in (**d**–**f**) or the corresponding isotype controls (light-colored curves). Panels (**a**,**b**,**d**,**e**) show histograms of cells transfected with the indicated siRNAs, (**c**) and (**f**) are the median fluorescence of the corresponding histograms shown in (**a**,**b**,**d**,**e**) (isotype controls subtracted). One typical result out of six experiments is shown.

**Figure 4 ijms-21-08770-f004:**
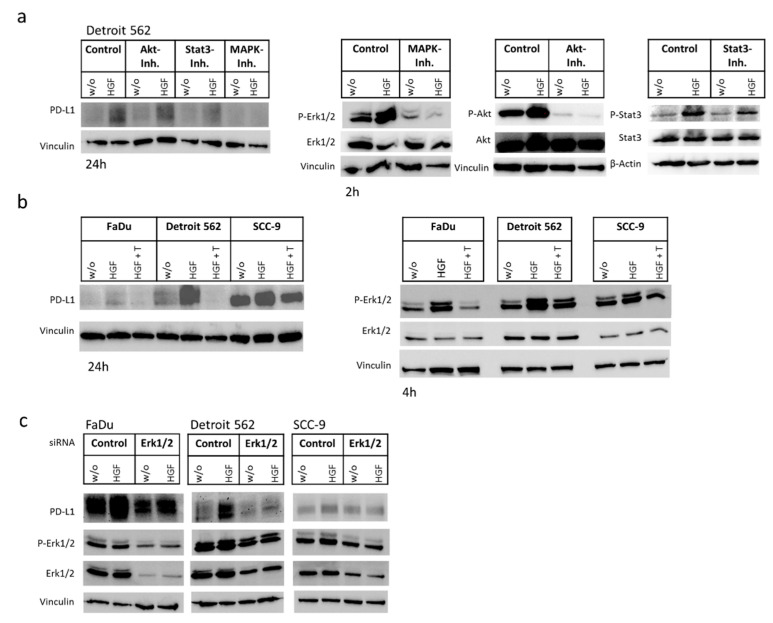
Increase in PD-L1 concentration is MAP kinase-dependent. (**a**) Lysates of Detroit 562 cells treated with 50 ng/mL HGF and/or the indicated inhibitors (Akt-Inh.: 1 µM MK-2206, Stat3-Inh.: 100 µM S3I-201, MAPK-Inh.: 1 µM SCH772984) for 24 h or corresponding untreated controls (*w*/*o*) were subjected to Western blotting using a PD-L1 specific antibody (on the left). To show the effect of HGF on the pathways and to check for the effectiveness of the inhibitors, samples were made after 2 h incubation as well, and a Western blot was performed with indicated antibodies (on the right). (**b**) Indicated cell lines were incubated for 24 h (on the left) or 4 h (on the right, to show the effect of HGF and Trametinib on pathway activation) with 50 ng/mL HGF alone or in combination with trametinib (T) or were left untreated (*w*/*o*). Effect on PD-L1 protein (on the left) and phosphorylated Erk1/2 protein (on the right) was investigated in a Western blot. Untreated samples served as controls (*w*/*o*). (**c**) Cells were transfected with Erk1/2 specific siRNA or a control siRNA. Two days later, cells were treated with 50 ng/mL HGF or remained untreated (*w*/*o*). After two more days, lysed cells were subjected to Western blotting using the indicated antibodies. Vinculin and β-actin were detected to check for equal gel loading. Typical results out of at least three experiments are shown.

**Figure 5 ijms-21-08770-f005:**
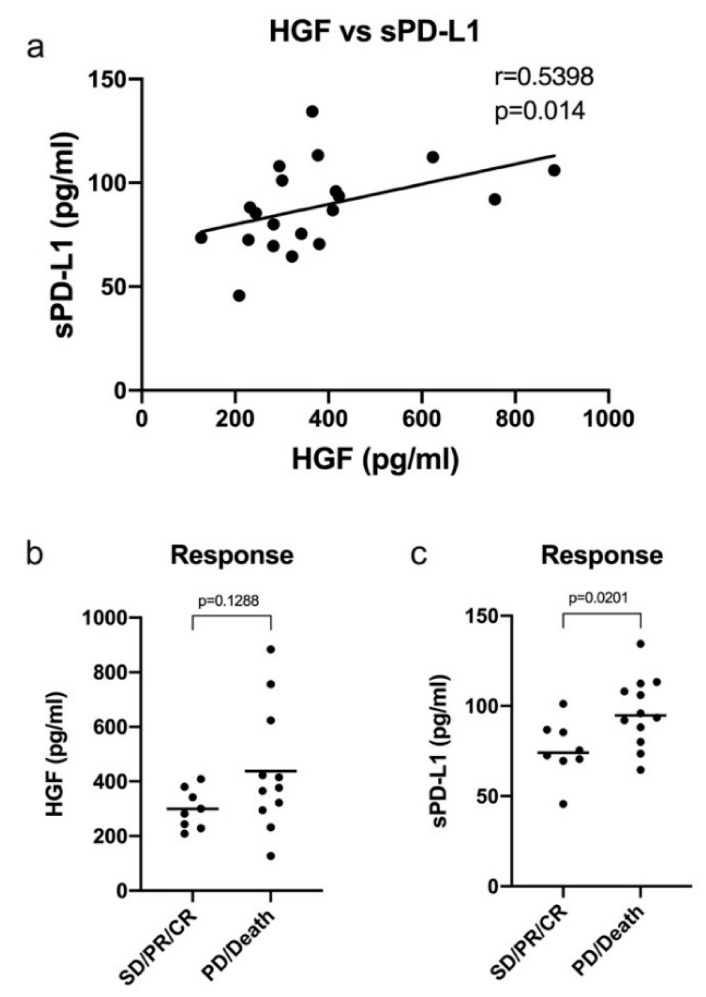
Serum levels of HGF and soluble programmed cell death protein 1 (sPD-L1) show a positive correlation in ICI treated HNSCC-patients. HGF and sPD-L1 ELISA results of serum from immune checkpoint inhibitor (ICI) treated patients with HNSCC (*n* = 20) were plotted on the x- and *y*-axis for correlation analysis (**a**). r: Spearman correlation coefficient, p: two-tailed *p*-value (α = 0.05). Panel (**b**,**c**) illustrate the same ELISA results as in (**a**) with respect to clinical response to ICI therapy. Mean HGF concentration in patients with stable disease (SD), partial remission (PR), or complete remission (CR) was 291.4 pg/mL. In non-responders, including progressive disease (PD) or death during therapy, mean HGF level was 371.3 pg/mL (**b**). Mean sPD-L1 level in patients responding to ICI was 74.02 pg/mL and 94.76 pg/mL in non-responders (**c**). p: two-tailed Mann–Whitney test (α = 0.05).

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
