# Peer review of "HGF-Induced PD-L1 Expression in Head and Neck Cancer: Preclinical and Clinical Findings"

_ijms, 2020, doi:10.3390/ijms21228770_

Round 1
Reviewer 1 Report
Thank you for the opportunity to review this interesting manuscript. The issue addressed is highly relevant, the manuscript is well structured and written appealingly. Also, the references to the results reported earlier by the authors is helpful to contextualize those new findings.
Some minor Points should be mentioned that may help to further improve the manuscript:
- Consider providing more detailed data on the Patient cohort including demographical, tumor specific and pathological data (maybe also some data on Prior Treatment and the Quality of Response to ICI).
- Please carefully revise the text for spelling mistakes and minor Grammar Errors.
Author Response
Dear Reviewer,
thank you very much for taking the time to review our manuscript. We tried to address the points you suggested in our new version:
- We included some more information on the patient cohort in the materials and methods section. Now additional information on the location of the primary tumor and the initial treatment of the patients is provided. As we used samples retrospectively (leftover material of another biomarker trial) we were not involved in the collection of the data. Also, the nature of HNSCC treatment results in inhomogeneous datasets regarding the pathological data, since some patients received surgical resection of the primary tumor and the regional lymph nodes whereas other patients are solely treated with chemoradiation. Unfortunately, the collected data are not homogeneous and therefore are not arranged in a meaningful way without going too much into detail.
- We spellchecked the new version of the article.
Thank you very much for your useful comments.
Best regards,
the authors
Reviewer 2 Report
The study describes the mechanisms involved in ICI response and resistance in HNSCC patients using tumor cell lines and patient serum samples. This is quite interesting to further anlyze these biomarkers in other cancers and potential comination therapies with Met inhibitors and ICI antibodies.
Results are clear and presented well for easy to understand.
The text requires some editing with English language expertise.
Figure 1 legend need to explain the time point 2 and 24 hours in 1A.
Author Response
Dear Reviewer,
thank you very much for taking the time to review our manuscript. See below what we changed to address the points you mentioned.
- We thoroughly checked the new version of our manuscript for grammatical and spelling errors.
- We reasoned that you meant the Western-blots showing different time points in figure 4a. The reason why we looked for the PD-L1 signal after 24 hours of stimulation and for the changes in signaling pathways already after 2 hours is, that the effect of HGF on the pathways is an early and transient event. The effect on PD-L1 on the other hand is a later effect involving new protein synthesis (increase in mRNA). We changed the figure legend accordingly for better understanding.
Thank you very much for your very useful comments.
Best regards,
the authors